# Stochastic Multi-Armed-Bandit Problem with Non-stationary Rewards

**Omar Besbes**
Columbia University
New York, NY
ob2105@columbia.edu

**Yonatan Gur**
Stanford University
Stanford, CA
ygur@stanford.edu

**Assaf Zeevi**
Columbia University
New York, NY
assaf@gsb.columbia.edu

## Abstract

In a multi-armed bandit (MAB) problem a gambler needs to choose at each round of play one of $K$ arms, each characterized by an unknown reward distribution. Reward realizations are only observed when an arm is selected, and the gambler's objective is to maximize his cumulative expected earnings over some given horizon of play $T$. To do this, the gambler needs to acquire information about arms (exploration) while simultaneously optimizing immediate rewards (exploitation); the price paid due to this trade off is often referred to as the *regret*, and the main question is how small can this price be as a function of the horizon length $T$. This problem has been studied extensively when the reward distributions do not change over time; an assumption that supports a sharp characterization of the regret, yet is often violated in practical settings. In this paper, we focus on a MAB formulation which allows for a broad range of temporal uncertainties in the rewards, while still maintaining mathematical tractability. We fully characterize the (regret) complexity of this class of MAB problems by establishing a direct link between the extent of allowable reward "variation" and the minimal achievable regret, and by establishing a connection between the adversarial and the stochastic MAB frameworks.

## 1 Introduction

**Background and motivation.** In the presence of uncertainty and partial feedback on rewards, an agent that faces a sequence of decisions needs to judiciously use information collected from past observations when trying to optimize future actions. A widely studied paradigm that captures this tension between the acquisition cost of new information (*exploration*) and the generation of instantaneous rewards based on the existing information (*exploitation*), is that of multi armed bandits (MAB), originally proposed in the context of drug testing by [1], and placed in a general setting by [2]. The original setting has a gambler choosing among $K$ slot machines at each round of play, and upon that selection observing a reward realization. In this classical formulation the rewards are assumed to be independent and identically distributed according to an unknown distribution that characterizes each machine. The objective is to maximize the expected sum of (possibly discounted) rewards received over a given (possibly infinite) time horizon. Since their inception, MAB problems with various modifications have been studied extensively in Statistics, Economics, Operations Research, and Computer Science, and are used to model a plethora of dynamic optimization problems under uncertainty; examples include clinical trials ([3]), strategic pricing ([4]), investment in innovation ([5]), packet routing ([6]), on-line auctions ([7]), assortment selection ([8]), and on-

line advertising ([9]), to name but a few. For overviews and further references cf. the monographs by [10], [11] for Bayesian / dynamic programming formulations, and [12] that covers the machine learning literature and the so-called adversarial setting. Since the set of MAB instances in which one can identify the optimal policy is extremely limited, a typical yardstick to measure performance of a candidate policy is to compare it to a benchmark: an *oracle* that at each time instant selects the arm that maximizes expected reward. The difference between the performance of the policy and that of the oracle is called the *regret*. When the growth of the regret as a function of the horizon $T$ is *sub-linear*, the policy is *long-run average optimal*: its long run average performance converges to that of the oracle. Hence the first order objective is to develop policies with this characteristic. The precise rate of growth of the regret as a function of $T$ provides a refined measure of policy performance. [13] is the first paper that provides a sharp characterization of the regret growth rate in the context of the traditional (stationary random rewards) setting, often referred to as the *stochastic* MAB problem. Most of the literature has followed this path with the objective of designing policies that exhibit the "slowest possible" rate of growth in the regret (often referred to as *rate optimal* policies).

In many application domains, several of which were noted above, temporal changes in the reward distribution structure are an intrinsic characteristic of the problem. These are ignored in the traditional stochastic MAB formulation, but there have been several attempts to extend that framework. The origin of this line of work can be traced back to [14] who considered a case where only the state of the chosen arm can change, giving rise to a rich line of work (see, e.g., [15], and [16]). In particular, [17] introduced the term *restless bandits*; a model in which the states (associated with reward distributions) of arms change in each step according to an arbitrary, yet known, stochastic process. Considered a hard class of problems (cf. [18]), this line of work has led to various approximations (see, e.g., [19]), relaxations (see, e.g., [20]), and considerations of more detailed processes (see, e.g., [21] for irreducible Markov process, and [22] for a class of history-dependent rewards).

Departure from the stationarity assumption that has dominated much of the MAB literature raises fundamental questions as to how one should model temporal uncertainty in rewards, and how to benchmark performance of candidate policies. One view, is to allow the reward realizations to be selected at any point in time by an *adversary*. These ideas have their origins in game theory with the work of [23] and [24], and have since seen significant development; [25] and [12] provide reviews of this line of research. Within this so called *adversarial* formulation, the efficacy of a policy over a given time horizon $T$ is often measured relative to a benchmark defined by the *single best action* one could have taken in hindsight (after seeing all reward realizations). The single best action benchmark represents a *static* oracle, as it is constrained to a single (static) action. This static oracle can perform quite poorly relative to a *dynamic oracle* that follows the optimal *dynamic* sequence of actions, as the latter optimizes the (expected) reward at each time instant over all possible actions.[1] Thus, a potential limitation of the adversarial framework is that even if a policy has a "small" regret relative to a static oracle, there is no guarantee with regard to its performance relative to the dynamic oracle.

**Main contributions.** The main contribution of this paper lies in fully characterizing the (regret) complexity of a broad class of MAB problems with non-stationary reward structure by establishing a direct link between the extent of reward "variation" and the minimal achievable regret. More specifically, the paper's contributions are along four dimensions. On the modeling side we formulate a class of non-stationary reward structure that is quite general, and hence can be used to realistically capture a variety of real-world type phenomena, yet is mathematically tractable. The main constraint that we impose on the evolution of the mean rewards is that their variation over the relevant time horizon is bounded by a *variation budget* $V_T$; a concept that was recently introduced in [26] in the context of non-stationary stochastic approximation. This limits the power of nature compared to the adversarial setup discussed above where rewards can be picked to maximally affect the policy's performance at each instance within $\{1, \ldots, T\}$. Nevertheless, this constraint allows for a rich class of temporal changes, extending most of the treatment in the non-stationary stochastic MAB literature, which mainly focuses on a finite number of changes in the mean rewards, see, e.g., [27] and references therein. We further discuss connections with studied non-stationary instances in §6.

The second dimension of contribution lies in the analysis domain. For a general class of non-stationary reward distributions we establish lower bounds on the performance of *any* non-anticipating policy relative to the *dynamic* oracle, and show that these bounds can be achieved,

uniformly over the class of admissible reward distributions, by a suitable policy construction. The term "achieved" is meant in the sense of the order of the regret as a function of the time horizon $T$, the variation budget $V_T$, and the number of arms $K$. Our policies are shown to be minimax optimal up to a term that is logarithmic in the number of arms, and the regret is sublinear and is of order $(KV_T)^{1/3} T^{2/3}$. Our analysis complements studied non-stationary instances by treating a broad and flexible class of temporal changes in the reward distributions, yet still establishing optimality results and showing that sublinear regret is achievable. Our results provide a spectrum of orders of the minimax regret ranging between order $T^{2/3}$ (when $V_T$ is a constant independent of $T$) and order $T$ (when $V_T$ grows linearly with $T$), mapping allowed variation to best achievable performance.

With the analysis described above we shed light on the exploration-exploitation trade off that characterizes the non-stationary reward setting, and the change in this trade off compared to the stationary setting. In particular, our results highlight the tension that exists between the need to "remember" and "forget." This is characteristic of several algorithms that have been developed in the adversarial MAB literature, e.g., the family of exponential weight methods such as EXP3, EXP3.S and the like; see, e.g., [28], and [12]. In a nutshell, the fewer past observations one retains, the larger the stochastic error associated with one's estimates of the mean rewards, while at the same time using more past observations increases the risk of these being biased.

One interesting observation drawn in this paper connects between the adversarial MAB setting, and the non-stationary environment studied here. In particular, as in [26], it is seen that an optimal policy in the adversarial setting may be suitably calibrated to perform near-optimally in the non-stationary stochastic setting. This will be further discussed after the main results are established.

## 2 Problem Formulation

Let $\mathcal{K} = \{1, \ldots, K\}$ be a set of arms. Let $\mathcal{T} = \{1, 2, \ldots, T\}$ denote a sequence of decision epochs faced by a decision maker. At any epoch $t \in \mathcal{T}$, the decision-maker pulls one of the $K$ arms. When pulling arm $k \in \mathcal{K}$ at epoch $t \in \mathcal{T}$, a reward $X_t^k \in [0, 1]$ is obtained, where $X_t^k$ is a random variable with expectation $\mu_t^k = \mathbb{E}\left[X_t^k\right]$. We denote the best possible expected reward at decision epoch $t$ by $\mu_t^*$, i.e., $\mu_t^* = \max_{k \in \mathcal{K}} \left\{\mu_t^k\right\}$.

**Changes in the expected rewards of the arms.** We assume the expected reward of each arm $\mu_t^k$ may change at any decision epoch. We denote by $\mu^k$ the sequence of expected rewards of arm $k$: $\mu^k = \left\{\mu_t^k\right\}_{t=1}^T$. In addition, we denote by $\mu$ the sequence of vectors of all $K$ expected rewards: $\mu = \left\{\mu^k\right\}_{k=1}^K$. We assume that the expected reward of each arm can change an arbitrary number of times, but bound the total variation of the expected rewards:

$$\sum_{t=1}^{T-1} \sup_{k \in \mathcal{K}} \left|\mu_t^k - \mu_{t+1}^k\right|. \tag{1}$$

Let $\{V_t : t = 1, 2, \ldots\}$ be a non-decreasing sequence of positive real numbers such that $V_1 = 0$, $KV_t \leq t$ for all $t$, and for normalization purposes set $V_2 = 2 \cdot K^{-1}$. We refer to $V_T$ as the *variation budget* over $\mathcal{T}$. We define the corresponding *temporal uncertainty set*, as the set of reward vector sequences that are subject to the variation budget $V_T$ over the set of decision epochs $\{1, \ldots, T\}$:

$$\mathcal{V} = \left\{\mu \in [0, 1]^{K \times T} : \sum_{t=1}^{T-1} \sup_{k \in \mathcal{K}} \left|\mu_t^k - \mu_{t+1}^k\right| \leq V_T\right\}.$$

The variation budget captures the constraint imposed on the non-stationary environment faced by the decision-maker. While limiting the possible evolution in the environment, it allows for numerous forms in which the expected rewards may change: continuously, in discrete shocks, and of a changing rate (Figure 1 depicts two different variation patterns that correspond to the same variation budget). In general, the variation budget $V_T$ is designed to depend on the number of pulls $T$.

**Admissible policies, performance, and regret.** Let $U$ be a random variable defined over a probability space $(\mathbb{U}, \mathcal{U}, \mathbf{P}_u)$. Let $\pi_1 : \mathbb{U} \to \mathcal{K}$ and $\pi_t : [0, 1]^{t-1} \times \mathbb{U} \to \mathcal{K}$ for $t = 2, 3, \ldots$ be measurable functions. With some abuse of notation we denote by $\pi_t \in \mathcal{K}$ the action at time $t$, that is given by

$$\pi_t = \begin{cases} \pi_1(U) & t = 1, \\ \pi_t\left(X_{t-1}^\pi, \ldots, X_1^\pi, U\right) & t = 2, 3, \ldots, \end{cases}$$

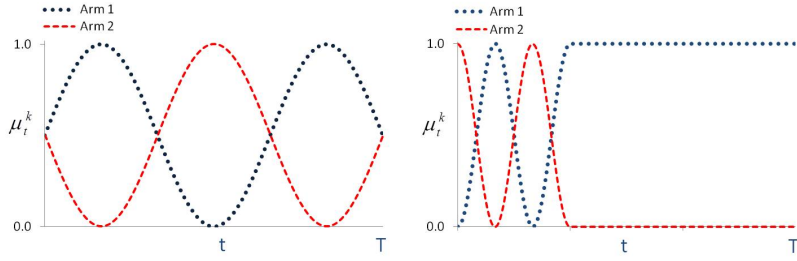

Figure 1: Two instances of variation in the mean rewards: (*Left*) A fixed variation budget (that equals 3) is "spent" over the whole horizon. (*Right*) The same budget is "spent" in the first third of the horizon.

The mappings $\{\pi_t : t = 1, \ldots, T\}$ together with the distribution $\mathbf{P}_u$ define the class of admissible policies. We denote this class by $\mathcal{P}$. We further denote by $\{\mathcal{H}_t, t = 1, \ldots, T\}$ the filtration associated with a policy $\pi \in \mathcal{P}$, such that $\mathcal{H}_1 = \sigma(U)$ and $\mathcal{H}_t = \sigma\left(\left\{X_j^\pi\right\}_{j=1}^{t-1}, U\right)$ for all $t \in \{2, 3, \ldots\}$. Note that policies in $\mathcal{P}$ are non-anticipating, i.e., depend only on the past history of actions and observations, and allow for randomized strategies via their dependence on $U$.

We define the *regret* under policy $\pi \in \mathcal{P}$ compared to a *dynamic* oracle as the worst-case difference between the expected performance of pulling at each epoch $t$ the arm which has the highest expected reward at epoch $t$ (the dynamic oracle performance) and the expected performance under policy $\pi$:

$$\mathcal{R}^\pi(\mathcal{V}, T) = \sup_{\mu \in \mathcal{V}} \left\{ \sum_{t=1}^T \mu_t^* - \mathbb{E}^\pi \left[ \sum_{t=1}^T \mu_t^\pi \right] \right\},$$

where the expectation $\mathbb{E}^\pi[\cdot]$ is taken with respect to the noisy rewards, as well as to the policy's actions. In addition, we denote by $\mathcal{R}^*(\mathcal{V}, T)$ the minimal worst-case regret that can be guaranteed by an admissible policy $\pi \in \mathcal{P}$, that is, $\mathcal{R}^*(\mathcal{V}, T) = \inf_{\pi \in \mathcal{P}} \mathcal{R}^\pi(\mathcal{V}, T)$. Then, $\mathcal{R}^*(\mathcal{V}, T)$ is the best achievable performance. In the following sections we study the magnitude of $\mathcal{R}^*(\mathcal{V}, T)$. We analyze the magnitude of this quantity by establishing upper and lower bounds; in these bounds we refer to a constant $C$ as *absolute* if it is independent of $K$, $V_T$, and $T$.

## 3   Lower bound on the best achievable performance

We next provide a lower bound on the the best achievable performance.

**Theorem 1** *Assume that rewards have a Bernoulli distribution. Then, there is some absolute constant $C > 0$ such that for any policy $\pi \in \mathcal{P}$ and for any $T \geq 1$, $K \geq 2$ and $V_T \in \left[K^{-1}, K^{-1}T\right]$,*

$$\mathcal{R}^\pi(\mathcal{V}, T) \geq C\left(KV_T\right)^{1/3} T^{2/3}.$$

We note that when reward distributions are stationary, there are known policies such as UCB1 ([29]) that achieve regret of order $\sqrt{T}$ in the stochastic setup. When the reward structure is non-stationary and defined by the class $\mathcal{V}$, then no policy may achieve such a performance and the best performance must incur a regret of at least order $T^{2/3}$. This additional complexity embedded in the non-stationary stochastic MAB problem compared to the stationary one will be further discussed in §6. We note that Theorem 1 also holds when $V_T$ is increasing with $T$. In particular, when the variation budget is linear in $T$, the regret grows linearly and long run average optimality is not achievable.

The driver of the change in the best achievable performance relative to the one established in a stationary environment, is a second tradeoff (over the tension between exploring different arms and capitalizing on the information already collected) introduced by the non-stationary environment, between "remembering" and "forgetting": estimating the expected rewards is done based on past observations of rewards. While keeping track of more observations may decrease the variance of mean rewards estimates, the non-stationary environment implies that "old" information is potentially less relevant due to possible changes in the underlying rewards. The changing rewards give incentive to dismiss old information, which in turn encourages enhanced exploration. The proof of Theorem 1 emphasizes the impact of these tradeoffs on the achievable performance.

**Key ideas in the proof.** At a high level the proof of Theorem 1 builds on ideas of identifying a worst-case "strategy" of nature (e.g., [28], proof of Theorem 5.1) adapting them to our setting. While the proof is deferred to the online companion (as supporting material), we next describe the key ideas when $V_T = 1$.[2] We define a subset of vector sequences $\mathcal{V}' \subset \mathcal{V}$ and show that when $\mu$ is drawn randomly from $\mathcal{V}'$, any admissible policy must incur regret of order $(KV_T)^{1/3} T^{2/3}$. We define a partition of the decision horizon $\mathcal{T}$ into batches $\mathcal{T}_1, \ldots, \mathcal{T}_m$ of size $\tilde{\Delta}_T$ each (except, possibly the last batch):

$$\mathcal{T}_j = \left\{ t \,:\, (j-1)\tilde{\Delta}_T + 1 \leq t \leq \min\left\{ j\tilde{\Delta}_T, T \right\} \right\}, \quad \text{for all } j = 1, \ldots, m, \tag{2}$$

where $m = \lceil T/\tilde{\Delta}_T \rceil$ is the number of batches. In $\mathcal{V}'$, in every batch there is exactly one "good" arm with expected reward $1/2 + \varepsilon$ for some $0 < \varepsilon \leq 1/4$, and all the other arms have expected reward $1/2$. The "good" arm is drawn independently in the beginning of each batch according to a discrete uniform distribution over $\{1, \ldots, K\}$. Thus, the identity of the "good" arm can change only between batches. By selecting $\varepsilon$ such that $\varepsilon T/\tilde{\Delta}_T \leq V_T$, any $\mu \in \mathcal{V}'$ is composed of expected reward sequences with a variation of at most $V_T$, and therefore $\mathcal{V}' \subset \mathcal{V}$. Given the draws under which expected reward sequences are generated, nature prevents any accumulation of information from one batch to another, since at the beginning of each batch a new "good" arm is drawn independently of the history. The proof of Theorem 1 establishes that when $\varepsilon \approx 1/\sqrt{\tilde{\Delta}_T}$ no admissible policy can identify the "good" arm with high probability within a batch. Since there are $\tilde{\Delta}_T$ epochs in each batch, the regret that any policy must incur along a batch is of order $\tilde{\Delta}_T \cdot \varepsilon \approx \sqrt{\tilde{\Delta}_T}$, which yields a regret of order $\sqrt{\tilde{\Delta}_T} \cdot T/\tilde{\Delta}_T \approx T/\sqrt{\tilde{\Delta}_T}$ throughout the whole horizon. Selecting the smallest feasible $\tilde{\Delta}_T$ such that the variation budget constraint is satisfied leads to $\tilde{\Delta}_T \approx T^{2/3}$, yielding a regret of order $T^{2/3}$ throughout the horizon.

## 4   A near-optimal policy

We apply the ideas underlying the lower bound in Theorem 1 to develop a rate optimal policy for the non-stationary stochastic MAB problem with a variation budget. Consider the following policy:

---

**Rexp3.** Inputs: a positive number $\gamma$, and a batch size $\Delta_T$.

1. Set batch index $j = 1$
2. Repeat while $j \leq \lceil T/\Delta_T \rceil$:
   (a) Set $\tau = (j-1)\Delta_T$
   (b) Initialization: for any $k \in \mathcal{K}$ set $w_t^k = 1$
   (c) Repeat for $t = \tau + 1, \ldots, \min\{T, \tau + \Delta_T\}$:
      - For each $k \in \mathcal{K}$, set

$$p_t^k = (1 - \gamma) \frac{w_t^k}{\sum_{k'=1}^K w_t^{k'}} + \frac{\gamma}{K}$$

      - Draw an arm $k'$ from $\mathcal{K}$ according to the distribution $\{p_t^k\}_{k=1}^K$
      - Receive a reward $X_t^{k'}$
      - For $k'$ set $\hat{X}_t^{k'} = X_t^{k'}/p_t^{k'}$, and for any $k \neq k'$ set $\hat{X}_t^k = 0$. For all $k \in \mathcal{K}$ update:

$$w_{t+1}^k = w_t^k \exp\left\{ \frac{\gamma \hat{X}_t^k}{K} \right\}$$

   (d) Set $j = j + 1$, and return to the beginning of step 2

---

Clearly $\pi \in \mathcal{P}$. The Rexp3 policy uses Exp3, a policy introduced by [30] for solving a worst-case sequential allocation problem, as a subroutine, restarting it every $\Delta_T$ epochs.

**Theorem 2** *Let $\pi$ be the Rexp3 policy with a batch size $\Delta_T = \left\lceil (K \log K)^{1/3} (T/V_T)^{2/3} \right\rceil$ and with $\gamma = \min \left\{ 1 \, , \, \sqrt{\frac{K \log K}{(e-1)\Delta_T}} \right\}$. Then, there is some absolute constant $\bar{C}$ such that for every $T \geq 1$, $K \geq 2$, and $V_T \in \left[ K^{-1}, K^{-1}T \right]$:*

$$\mathcal{R}^\pi(\mathcal{V}, T) \leq \bar{C} \left( K \log K \cdot V_T \right)^{1/3} T^{2/3}.$$

Theorem 2 is obtained by establishing a connection between the regret relative to the single best action in the adversarial setting, and the regret with respect to the dynamic oracle in non-stationary stochastic setting with variation budget. Several classes of policies, such as exponential-weight (including Exp3) and polynomial-weight policies, have been shown to achieve regret of order $\sqrt{T}$ with respect to the single best action in the adversarial setting (see chapter 6 of [12] for a review). While in general these policies tend to perform well numerically, there is no guarantee for their performance relative to the *dynamic oracle* studied in this paper, since the single best action *itself* may incur linear regret relative to the dynamic oracle; see also [31] for a study of the empirical performance of one class of algorithms. The proof of Theorem 2 shows that *any* policy that achieves regret of order $\sqrt{T}$ with respect to the single best action in the adversarial setting, can be used as a subroutine to obtain near-optimal performance with respect to the dynamic oracle in our setting.

Rexp3 emphasizes the two tradeoffs discussed in the previous section. The first tradeoff, information acquisition versus capitalizing on existing information, is captured by the subroutine policy Exp3. In fact, any policy that achieves a good performance compared to a single best action benchmark in the adversarial setting must balance exploration and exploitation. The second tradeoff, "remembering" versus "forgetting," is captured by restarting Exp3 and forgetting any acquired information every $\Delta_T$ pulls. Thus, old information that may slow down the adaptation to the changing environment is being discarded. Theorem 1 and Theorem 2 together characterize the minimax regret (up to a multiplicative factor, logarithmic in the number of arms) in a full spectrum of variations $V_T$:

$$\mathcal{R}^*(\mathcal{V}, T) \asymp (KV_T)^{1/3} T^{2/3}.$$

Hence, we have quantified the impact of the extent of change in the environment on the best achievable performance in this broad class of problems. For example, for the case in which $V_T = C \cdot T^\beta$, for some absolute constant $C$ and $0 \leq \beta < 1$ the best achievable regret is of order $T^{(2+\beta)/3}$.

We finally note that restarting is only one way of adapting policies from the adversarial MAB setting to achieve near optimality in the non-stationary stochastic setting; a way that articulates well the principles leading to near optimality. In the online companion we demonstrate that near optimality can be achieved by other adaptation methods, showing that the Exp3.S policy (given in [28]) can be tuned by $\alpha = \frac{1}{T}$ and $\gamma \approx (KV_T/T)^{1/3}$ to achieve near optimality in our setting, without restarting.

## 5 Proof of Theorem 2

The structure of the proof is as follows. First, we break the horizon to a sequence of batches of size $\Delta_T$ each, and analyze the performance gap between the single best action and the dynamic oracle in each batch. Then, we plug in a known performance guarantee for Exp3 relative to the single best action, and sum over batches to establish the regret of Rexp3 relative to the dynamic oracle.

**Step 1 (Preliminaries).** Fix $T \geq 1$, $K \geq 2$, and $V_T \in \left[ K^{-1}, K^{-1}T \right]$. Let $\pi$ be the Rexp3 policy, tuned by $\gamma = \min \left\{ 1 \, , \, \sqrt{\frac{K \log K}{(e-1)\Delta_T}} \right\}$ and $\Delta_T \in \{1, \ldots, T\}$ (to be specified later on). We break the horizon $\mathcal{T}$ into a sequence of batches $\mathcal{T}_1, \ldots, \mathcal{T}_m$ of size $\Delta_T$ each (except, possibly $\mathcal{T}_m$) according to (2). Let $\mu \in \mathcal{V}$, and fix $j \in \{1, \ldots, m\}$. We decomposition the regret in batch $j$:

$$\mathbb{E}^\pi \left[ \sum_{t \in \mathcal{T}_j} (\mu_t^* - \mu_t^\pi) \right] = \underbrace{\sum_{t \in \mathcal{T}_j} \mu_t^* - \mathbb{E} \left[ \max_{k \in \mathcal{K}} \left\{ \sum_{t \in \mathcal{T}_j} X_t^k \right\} \right]}_{J_{1,j}} + \underbrace{\mathbb{E} \left[ \max_{k \in \mathcal{K}} \left\{ \sum_{t \in \mathcal{T}_j} X_t^k \right\} \right] - \mathbb{E}^\pi \left[ \sum_{t \in \mathcal{T}_j} \mu_t^\pi \right]}_{J_{2,j}}.$$

(3)

The first component, $J_{1,j}$, is the expected loss associated with using a single action over batch $j$. The second component, $J_{2,j}$, is the expected regret relative to the best static action in batch $j$.

**Step 2 (Analysis of $J_{1,j}$ and $J_{2,j}$).** Defining $\mu^k_{T+1} = \mu^k_T$ for all $k \in \mathcal{K}$, we denote the variation in expected rewards along batch $\mathcal{T}_j$ by $V_j = \sum_{t \in \mathcal{T}_j} \max_{k \in \mathcal{K}} \left|\mu^k_{t+1} - \mu^k_t\right|$. We note that:

$$\sum_{j=1}^m V_j = \sum_{j=1}^m \sum_{t \in \mathcal{T}_j} \max_{k \in \mathcal{K}} \left|\mu^k_{t+1} - \mu^k_t\right| \leq V_T. \tag{4}$$

Let $k_0$ be an arm with best expected performance over $\mathcal{T}_j$: $k_0 \in \arg\max_{k \in \mathcal{K}} \left\{\sum_{t \in \mathcal{T}_j} \mu^k_t\right\}$. Then,

$$\max_{k \in \mathcal{K}} \left\{\sum_{t \in \mathcal{T}_j} \mu^k_t\right\} = \sum_{t \in \mathcal{T}_j} \mu^{k_0}_t = \mathbb{E}\left[\sum_{t \in \mathcal{T}_j} X^{k_0}_t\right] \leq \mathbb{E}\left[\max_{k \in \mathcal{K}} \left\{\sum_{t \in \mathcal{T}_j} X^k_t\right\}\right], \tag{5}$$

and therefore, one has:

$$\begin{aligned} J_{1,j} &= \sum_{t \in \mathcal{T}_j} \mu^*_t - \mathbb{E}\left[\max_{k \in \mathcal{K}} \left\{\sum_{t \in \mathcal{T}_j} X^k_t\right\}\right] \overset{(a)}{\leq} \sum_{t \in \mathcal{T}_j} \left(\mu^*_t - \mu^{k_0}_t\right) \\ &\leq \Delta_T \max_{t \in \mathcal{T}_j} \left\{\mu^*_t - \mu^{k_0}_t\right\} \overset{(b)}{\leq} 2V_j \Delta_T, \end{aligned} \tag{6}$$

for any $\mu \in \mathcal{V}$ and $j \in \{1, \dots, m\}$, where $(a)$ holds by (5) and $(b)$ holds by the following argument: otherwise there is an epoch $t_0 \in \mathcal{T}_j$ for which $\mu^*_{t_0} - \mu^{k_0}_{t_0} > 2V_j$. Indeed, let $k_1 = \arg\max_{k \in \mathcal{K}} \mu^k_{t_0}$. In such case, for all $t \in \mathcal{T}_j$ one has $\mu^{k_1}_t \geq \mu^{k_1}_{t_0} - V_j > \mu^{k_0}_{t_0} + V_j \geq \mu^{k_0}_t$, since $V_j$ is the maximal variation in batch $\mathcal{T}_j$. This however, contradicts the optimality of $k_0$ at epoch $t$, and thus (6) holds.

In addition, Corollary 3.2 in [28] points out that the regret incurred by Exp3 (tuned by $\gamma = \min\left\{1, \sqrt{\frac{K \log K}{(e-1)\Delta_T}}\right\}$) along $\Delta_T$ batches, relative to the single best action, is bounded by $2\sqrt{e-1}\sqrt{\Delta_T K \log K}$. Therefore, for each $j \in \{1, \dots, m\}$ one has

$$J_{2,j} = \mathbb{E}\left[\max_{k \in \mathcal{K}} \left\{\sum_{t \in \mathcal{T}_j} X^k_t\right\} - \mathbb{E}^\pi\left[\sum_{t \in \mathcal{T}_j} \mu^\pi_t\right]\right] \overset{(a)}{\leq} 2\sqrt{e-1}\sqrt{\Delta_T K \log K}, \tag{7}$$

for any $\mu \in \mathcal{V}$, where $(a)$ holds since within each batch arms are pulled according to Exp3($\gamma$).

**Step 3 (Regret throughout the horizon).** Summing over $m = \lceil T/\Delta_T \rceil$ batches we have:

$$\begin{aligned} \mathcal{R}^\pi(\mathcal{V}, T) &= \sup_{\mu \in \mathcal{V}} \left\{\sum_{t=1}^T \mu^*_t - \mathbb{E}^\pi\left[\sum_{t=1}^T \mu^\pi_t\right]\right\} \overset{(a)}{\leq} \sum_{j=1}^m \left(2\sqrt{e-1}\sqrt{\Delta_T K \log K} + 2V_j \Delta_T\right) \\ &\overset{(b)}{\leq} \left(\frac{T}{\Delta_T} + 1\right) \cdot 2\sqrt{e-1}\sqrt{\Delta_T K \log K} + 2\Delta_T V_T. \\ &= \frac{2\sqrt{e-1}\sqrt{K \log K} \cdot T}{\sqrt{\Delta_T}} + 2\sqrt{e-1}\sqrt{\Delta_T K \log K} + 2\Delta_T V_T, \end{aligned} \tag{8}$$

where: (a) holds by (3), (6), and (7); and (b) follows from (4). Finally, selecting $\Delta_T = \left\lceil (K \log K)^{1/3} (T/V_T)^{2/3}\right\rceil$, we establish:

$$\begin{aligned} \mathcal{R}^\pi(\mathcal{V}, T) &\leq 2\sqrt{e-1}\,(K \log K \cdot V_T)^{1/3}\, T^{2/3} \\ &\quad + 2\sqrt{e-1}\sqrt{\left((K \log K)^{1/3} (T/V_T)^{2/3} + 1\right) K \log K} \\ &\quad + 2\left((K \log K)^{1/3} (T/V_T)^{2/3} + 1\right) V_T \\ &\overset{(a)}{\leq} \left(\left(2 + 2\sqrt{2}\right)\sqrt{e-1} + 4\right)(K \log K \cdot V_T)^{1/3}\, T^{2/3}, \end{aligned}$$

where (a) follows from $T \geq K \geq 2$, and $V_T \in \left[K^{-1}, K^{-1}T\right]$. This concludes the proof. ∎

# 6 Discussion

**Unknown variation budget.** The Rexp3 policy relies on prior knowledge of $V_T$, but predictions of $V_T$ may be inaccurate (such estimation can be maintained from historical data if actions are occasionally randomized, for example, by fitting $V_T = T^\alpha$). Denoting the "true" variation budget by $V_T$ and the estimate that is used by the agent when tuning Rexp3 by $\hat{V}_T$, one may observe that the analysis in the proof of Theorem 2 holds until equation (8), but then $\Delta_T$ will be tuned using $\hat{V}_T$. This implies that when $V_T$ and $\hat{V}_T$ are "close," Rexp3 still guarantees long-run average optimality. For example, suppose that Rexp3 is tuned by $\hat{V}_T = T^\alpha$, but the variation is $V_T = T^{\alpha+\delta}$. Then sublinear regret (of order $T^{2/3+\alpha/3+\delta}$) is guaranteed as long as $\delta < (1-\alpha)/3$; e.g., if $\alpha = 0$ and $\delta = 1/4$, Rexp3 guarantees regret of order $T^{11/12}$ (accurate tuning would have guaranteed order $T^{3/4}$). Since there are no restrictions on the rate at which the variation budget can be spent, an interesting and potentially challenging open problem is to delineate to what extent it is possible to design adaptive policies that do not use prior knowledge of $V_T$, yet guarantee "good" performance.

**Contrasting with traditional (stationary) MAB problems.** The characterized minimax regret in the stationary stochastic setting is of order $\sqrt{T}$ when expected rewards can be arbitrarily close to each other, and of order $\log T$ when rewards are "well separated" (see [13] and [29]). Contrasting the minimax regret (of order $V_T^{1/3}T^{2/3}$) we have established in the stochastic non-stationary MAB problem with those established in stationary settings allows one to quantify the "price of non-stationarity," which mathematically captures the added complexity embedded in changing rewards versus stationary ones (as a function of the allowed variation). Clearly, additional complexity is introduced even when the allowed variation is fixed and independent of the horizon length.

**Contrasting with other non-stationary MAB instances.** The class of MAB problems with non-stationary rewards that is formulated in the current chapter extends other MAB formulations that allow rewards to change in a more structured manner. For example, [32] consider a setting where rewards evolve according to a Brownian motion and regret is linear in $T$; our results (when $V_T$ is linear in $T$) are consistent with theirs. Two other representative studies are those of [27], that study a stochastic MAB problems in which expected rewards may change a finite number of times, and [28] that formulate an adversarial MAB problem in which the identity of the best arm may change a finite number of times. Both studies suggest policies that, utilizing the prior knowledge that the number of changes must be finite, achieve regret of order $\sqrt{T}$ relative to the best sequence of actions. However, the performance of these policies can deteriorate to regret that is linear in $T$ when the number of changes is allowed to depend on $T$. When there is a finite variation ($V_T$ is fixed and independent of $T$) but not necessarily a finite number of changes, we establish that the best achievable performance deteriorate to regret of order $T^{2/3}$. In that respect, it is not surprising that the "hard case" used to establish the lower bound in Theorem 1 describes a nature's strategy that allocates variation over a large (as a function of $T$) number of changes in the expected rewards.

**Low variation rates.** While our formulation focuses on "significant" variation in the mean rewards, our established bounds also hold for "smaller" variation scales; when $V_T$ decreases from $O(1)$ to $O(T^{-1/2})$ the minimax regret rate decreases from $T^{2/3}$ to $\sqrt{T}$. Indeed, when the variation scale is $O(T^{-1/2})$ or smaller, the rate of regret coincides with that of the classical stochastic MAB setting.

## Footnotes

[1]Under non-stationary rewards it is immediate that the single best action may be sub-optimal in many decision epochs, and the performance gap between the static and the dynamic oracles can grow linearly with $T$.

[2]For the sake of simplicity, the discussion in this paragraph assumes a variation budget that is fixed and independent of $T$; the proof of Theorem 3 details a general treatment for a budget that depends on $T$.

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
