[Supplementary Material]

# Supporting Material:

# Stochastic Multi-Armed-Bandit Problem with Non-stationary Rewards

**Omar Besbes**
Columbia University
New York, NY
ob2105@columbia.edu

**Yonatan Gur**
Columbia University
New York, NY
ygur14@gsb.columbia.edu

**Assaf Zeevi**
Columbia University
New York, NY
assaf@gsb.columbia.edu

## A   Proof of Theorem 1

At a high level the proof adapts a general approach of identifying a worst-case nature "strategy" (see proof of Theorem 5.1 in [1], which analyze the worst-case regret relative to a single best action benchmark in a fully adversarial environment), extending these ideas appropriately to our setting. Fix $T \geq 1$, $K \geq 2$, and $V_T \in \left[ K^{-1}, K^{-1}T \right]$. In what follows we restrict nature to the class $\mathcal{V}' \subseteq \mathcal{V}$ that was described in §3, and show that when $\mu$ is drawn randomly from $\mathcal{V}'$, any policy in $\mathcal{P}$ must incur regret of order $(KV_T)^{1/3} \, T^{2/3}$.

**Step 1 (Preliminaries).** Define a partition of the decision horizon $\mathcal{T}$ to $m = \left\lceil \frac{T}{\tilde{\Delta}_T} \right\rceil$ batches $\mathcal{T}_1, \ldots, \mathcal{T}_m$ batches of size $\tilde{\Delta}_T$ each (except perhaps $\mathcal{T}_m$) according to Equation (2) in the main text. For some $\varepsilon > 0$ that will be specified shortly, define $\mathcal{V}'$ to be the set of reward vectors sequences $\mu$ such that:

- $\mu_t^k \in \{1/2, 1/2 + \varepsilon\}$ for all $k \in \mathcal{K}$, $t \in \mathcal{T}$
- $\sum_{k \in \mathcal{K}} \mu_t^k = K/2 + \varepsilon$ for all $t \in \mathcal{T}$
- $\mu_t^k = \mu_{t+1}^k$ for any $(j-1)\tilde{\Delta}_T + 1 \leq t \leq \min\left\{ j\tilde{\Delta}_T, T \right\} - 1$, $j = 1, \ldots, m$, for all $k \in \mathcal{K}$

For each sequence in $\mathcal{V}'$ in any epoch there is exactly one arm with expected reward $1/2 + \varepsilon$ where the rest of the arms have expected reward $1/2$, and expected rewards cannot change within a batch. Let $\varepsilon = \min\left\{ \frac{1}{4} \cdot \sqrt{K/\tilde{\Delta}_T}, V_T \tilde{\Delta}_T / T \right\}$. Then, for any $\mu \in \mathcal{V}'$ one has:

$$\sum_{t=1}^{T-1} \sup_{k \in \mathcal{K}} \left| \mu_t^k - \mu_{t+1}^k \right| \ \leq \ \sum_{j=1}^{m-1} \varepsilon \ = \ \left( \left\lceil \frac{T}{\tilde{\Delta}_T} \right\rceil - 1 \right) \cdot \varepsilon \ \leq \ \frac{T\varepsilon}{\tilde{\Delta}_T} \ \leq \ V_T,$$

where the first inequality follows from the structure of $\mathcal{V}'$. Therefore, $\mathcal{V}' \subset \mathcal{V}$.

**Step 2 (Single batch analysis).** Fix some policy $\pi \in \mathcal{P}$, and fix a batch $j \in \{1, \ldots, m\}$. Let $k_j$ denote the "good" arm of batch $j$. We denote by $\mathbb{P}_{k_j}^j$ the probability distribution conditioned on arm $k_j$ being the "good" arm in batch $j$, and by $\mathbb{P}_0$ the probability distribution with respect to

random rewards (i.e. expected reward $1/2$) for each arm. We further denote by $\mathbb{E}_{k_j}^j[\cdot]$ and $\mathbb{E}_0[\cdot]$ the respective expectations. Assuming binary rewards, we let $X$ denote a vector of $|\mathcal{T}_j|$ rewards, i.e. $X \in \{0,1\}^{|\mathcal{T}_j|}$. We denote by $N_k^j$ the number of times arm $k$ was selected in batch $j$. In the proof we use Lemma A.1 from [1] that characterizes the difference between the two different expectations of some function of the observed rewards vector:

**Lemma 1** *Let $f : \{0,1\}^{|\mathcal{T}_j|} \to [0, M]$ be a bounded real function. Then, for any $k \in \mathcal{K}$:*

$$\mathbb{E}_k^j[f(X)] - \mathbb{E}_0[f(X)] \leq \frac{M}{2}\sqrt{-\mathbb{E}_0\left[N_k^j\right]\log\left(1 - 4\varepsilon^2\right)}.$$

Recalling that $k_j$ denotes the "good" arm of batch $j$, one has

$$\mathbb{E}_{k_j}^j[\mu_t^\pi] = \left(\frac{1}{2} + \varepsilon\right)\mathbb{P}_{k_j}^j\{\pi_t = k_j\} + \frac{1}{2}\mathbb{P}_{k_j}^j\{\pi_t \neq k_j\} = \frac{1}{2} + \varepsilon\mathbb{P}_{k_j}^j\{\pi_t = k_j\},$$

and therefore,

$$\mathbb{E}_{k_j}^j\left[\sum_{t \in \mathcal{T}_j}\mu_t^\pi\right] = \frac{|\mathcal{T}_j|}{2} + \sum_{t \in \mathcal{T}_j}\varepsilon\mathbb{P}_{k_j}^j\{\pi_t = k_j\} = \frac{|\mathcal{T}_j|}{2} + \varepsilon\mathbb{E}_{k_j}^j\left[N_{k_j}^j\right]. \tag{1}$$

In addition, applying Lemma 1 with $f(X) = N_{k_j}^j$ (clearly $N_{k_j}^j \in \{0, \dots, |\mathcal{T}_j|\}$) we have:

$$\mathbb{E}_{k_j}^j\left[N_{k_j}^j\right] \leq \mathbb{E}_0\left[N_{k_j}^j\right] + \frac{|\mathcal{T}_j|}{2}\sqrt{-\mathbb{E}_0\left[N_{k_j}^j\right]\log\left(1 - 4\varepsilon^2\right)}.$$

Summing over arms, one has:

$$\begin{aligned}
\sum_{k_j=1}^K \mathbb{E}_{k_j}^j\left[N_{k_j}^j\right] &\leq \sum_{k_j=1}^K \mathbb{E}_0\left[N_{k_j}^j\right] + \sum_{k_j=1}^K \frac{|\mathcal{T}_j|}{2}\sqrt{-\mathbb{E}_0\left[N_{k_j}^j\right]\log\left(1 - 4\varepsilon^2\right)} \\
&\leq |\mathcal{T}_j| + \frac{|\mathcal{T}_j|}{2}\sqrt{-\log\left(1 - 4\varepsilon^2\right)|\mathcal{T}_j|\,K},
\end{aligned} \tag{2}$$

for any $j \in \{1, \dots, m\}$, where the last inequality holds since $\sum_{k_j=1}^K \mathbb{E}_0\left[N_{k_j}^j\right] = |\mathcal{T}_j|$, and thus by Cauchy-Schwarz inequality $\sum_{k_j=1}^K \sqrt{\mathbb{E}_0\left[N_{k_j}^j\right]} \leq \sqrt{|\mathcal{T}_j|\,K}$.

**Step 3 (Regret along the horizon).** Let $\tilde{\mu}$ be a random sequence of expected rewards vectors, in which in every batch the "good" arm is drawn according to an independent uniform distribution over the set $\mathcal{K}$. Clearly, every realization of $\tilde{\mu}$ is in $\mathcal{V}'$. In particular, taking expectation over $\tilde{\mu}$, one has:

$$\begin{aligned}
\mathcal{R}^\pi(\mathcal{V}', T) &= \sup_{\mu \in \mathcal{V}'}\left\{\sum_{t=1}^T \mu_t^* - \mathbb{E}^\pi\left[\sum_{t=1}^T \mu_t^\pi\right]\right\} \geq \mathbb{E}^{\tilde{\mu}}\left[\sum_{t=1}^T \tilde{\mu}_t^* - \mathbb{E}^\pi\left[\sum_{t=1}^T \tilde{\mu}_t^\pi\right]\right] \\
&\geq \sum_{j=1}^m\left(\sum_{t \in \mathcal{T}_j}\left(\frac{1}{2} + \varepsilon\right) - \frac{1}{K}\sum_{k_j=1}^K \mathbb{E}^\pi\mathbb{E}_{k_j}^j\left[\sum_{t \in \mathcal{T}_j}\tilde{\mu}_t^\pi\right]\right) \\
&\overset{(a)}{\geq} \sum_{j=1}^m\left(\sum_{t \in \mathcal{T}_j}\left(\frac{1}{2} + \varepsilon\right) - \frac{1}{K}\sum_{k_j=1}^K\left(\frac{|\mathcal{T}_j|}{2} + \varepsilon\mathbb{E}^\pi\mathbb{E}_{k_j}^j\left[N_{k_j}^j\right]\right)\right) \\
&\geq \sum_{j=1}^m\left(\sum_{t \in \mathcal{T}_j}\left(\frac{1}{2} + \varepsilon\right) - \frac{|\mathcal{T}_j|}{2} - \frac{\varepsilon}{K}\mathbb{E}^\pi\sum_{k_j=1}^K\mathbb{E}_{k_j}^j\left[N_{k_j}^j\right]\right) \\
&\overset{(b)}{\geq} T\varepsilon - \frac{T\varepsilon}{K} - \frac{T\varepsilon}{2K}\sqrt{-\log\left(1 - 4\varepsilon^2\right)\tilde{\Delta}_T K} \\
&\overset{(c)}{\geq} \frac{T\varepsilon}{2} - \frac{T\varepsilon^2}{K}\sqrt{\log\left(4/3\right)\tilde{\Delta}_T K},
\end{aligned}$$

where: $(a)$ holds by (1); $(b)$ holds by (2), since $\sum_{j=1}^{m} |\mathcal{T}_j| = T$, since $m \geq T/\tilde{\Delta}_T$, and since $|\mathcal{T}_j| \leq \tilde{\Delta}_T$ for all $j \in \{1, \ldots, m\}$; and $(c)$ holds by $4\varepsilon^2 \leq 1/4$, and $-\log(1-x) \leq 4\log(4/3)x$ for all $x \in [0, 1/4]$, and since $K \geq 2$. Set $\tilde{\Delta}_T = \left\lceil K^{1/3} \left(\frac{T}{V_T}\right)^{2/3} \right\rceil$. Recall that $\varepsilon = \min\left\{\frac{1}{4} \cdot \sqrt{K/\tilde{\Delta}_T}, V_T\tilde{\Delta}_T/T\right\}$. Then, one has:

$$
\begin{aligned}
\mathcal{R}^\pi(\mathcal{V}', T) &\geq T\varepsilon\left(\frac{1}{2} - \varepsilon\sqrt{\frac{\tilde{\Delta}_T\log(4/3)}{K}}\right) \geq T\varepsilon\left(\frac{1}{2} - \frac{\sqrt{\log(4/3)}}{4}\right) \\
&\geq \frac{1}{4}\cdot\min\left\{\frac{T}{4}\cdot\sqrt{\frac{K}{\tilde{\Delta}_T}}, V_T\tilde{\Delta}_T\right\} \\
&\geq \frac{1}{4}\cdot\min\left\{\frac{T}{4}\cdot\sqrt{\frac{K}{2K^{1/3}(T/V_T)^{2/3}}}, (KV_T)^{1/3}T^{2/3}\right\} \\
&\geq \frac{1}{4\sqrt{2}}\cdot(KV_T)^{1/3}T^{2/3}.
\end{aligned}
$$

This concludes the proof. $\blacksquare$

## B  Continuous updating

In this section we show that near optimality in the non-stationary stochastic setting can be obtained by re-tuning the Exp3.S policy, introduced in [1]:

---

**Exp3.S.** Inputs: a positive numbers $\gamma$, and $\alpha$.

1. Initialization: for any $k \in \mathcal{K}$ set $w_t^k = 1$
2. For each $t = 1, 2, \ldots$:
   - For each $k \in \mathcal{K}$, set
     $$p_t^k = (1-\gamma)\frac{w_t^k}{\sum_{k'=1}^{K} w_t^{k'}} + \frac{\gamma}{K}$$
   - Draw an arm $k'$ from $\mathcal{K}$ according to the distribution $\{p_t^k\}_{k=1}^{K}$
   - Receive a reward $X_t^{k'}$
   - For $k'$ set $\hat{X}_t^{k'} = X_t^{k'}/p_t^{k'}$, and for any $k \neq k'$ set $\hat{X}_t^k = 0$. For all $k \in \mathcal{K}$ update:
     $$w_{t+1}^k = w_t^k\exp\left\{\frac{\gamma\hat{X}_t^k}{K}\right\} + \frac{e\alpha}{K}\sum_{k'=1}^{K} w_{k'}(t)$$

---

We next prove that by selecting the tuning parameters to be $\alpha = \frac{1}{T}$ and $\gamma = \min\left\{1, \left(\frac{2V_T K\log(KT)}{(e-1)^2 T}\right)^{1/3}\right\}$, Exp3.S achieve near optimal performance is the non-stationary stochastic setting. The structure of the proof is follows: First, we follow the proof of Theorem 2 (see main text), breaking the decision horizon to a sequence of decision batches and analyzing the difference in performance between the sequence of single best actions and the performance of the dynamic oracle. Then, we analyze the regret of the Exp3.S policy when compared to the sequence composed of the single best actions of each batch (this part of the proof roughly follows the proof lines of Theorem 8.1 of [1], while considering a possibly infinite number of changes in the identity of the best arm). Finally, we select tuning parameters that minimize the overall regret.

**Step 1 (Preliminaries).** Fix $T \geq 1$, $K \geq 2$, and $TK^{-1} \geq V_T \geq K^{-1}$. Let $\pi$ be the Exp3.S policy (the tuning parameters with be set later). We break the decision horizon $\mathcal{T}$ to batches $\mathcal{T}_1, \ldots, \mathcal{T}_m$ of size $\Delta_T$ each (except perhaps $\mathcal{T}_m$) according to step 1 in the proof of Theorem 2 (see main text).

**Step 2.** Let $\mu \in \mathcal{V}$. We follow the proof of Theorem 2 (see the beginning of step 3) to obtain:

$$\mathbb{E}^{\pi}\left[\sum_{t\in\mathcal{T}_j}(\mu_t^* - \mu_t^{\pi})\right] = \sum_{t\in\mathcal{T}_j}\mu_t^* - \max_{k\in\mathcal{K}}\left\{\sum_{t\in\mathcal{T}_j}\mu_t^k\right\} + \max_{k\in\mathcal{K}}\left\{\sum_{t\in\mathcal{T}_j}\mu_t^k\right\} - \mathbb{E}^{\pi}\left[\sum_{t\in\mathcal{T}_j}\mu_t^{\pi}\right]$$

$$\leq 2V_j\Delta_T + \max_{k\in\mathcal{K}}\left\{\sum_{t\in\mathcal{T}_j}\mu_t^k\right\} - \mathbb{E}^{\pi}\left[\sum_{t\in\mathcal{T}_j}\mu_t^{\pi}\right], \tag{3}$$

for each $j \in \{1,\ldots,m\}$ and for any $\mu \in \mathcal{V}$. Fix $j \in \{1,\ldots,m\}$. We next bound the difference between the performance of the single best action in $\mathcal{T}_j$ and that of the policy, throughout $\mathcal{T}_j$. Let $t_j$ denote the first decision index of batch $j$, that is, $t_j = (j-1)\Delta_T + 1$. We $W_t$ denote the sum of all weights at decision $t$: $W_t = \sum_{k=1}^K w_t^k$. Following the proof of Theorem 8.1 in [1], one has:

$$\frac{W_{t+1}}{W_t} \leq 1 + \frac{\gamma/K}{1-\gamma}X_t^{\pi} + \frac{(e-2)(\gamma/K)^2}{1-\gamma}\sum_{k=1}^K \hat{X}_t^k + e\alpha. \tag{4}$$

Taking logarithms on both sides of (4) and summing over all $t \in \mathcal{T}_j$, we get:

$$\log\left(\frac{W_{t_{j+1}}}{W_{t_j}}\right) \leq \frac{\gamma/K}{1-\gamma}\sum_{t\in\mathcal{T}_j}X_t^{\pi} + \frac{(e-2)(\gamma/K)^2}{1-\gamma}\sum_{t\in\mathcal{T}_j}\sum_{k=1}^K \hat{X}_t^k + e\alpha\,|\mathcal{T}_j|, \tag{5}$$

where for $\mathcal{T}_m$ set $W_{t_{m+1}} = W_T$. Let $k_j$ be the best single action in $\mathcal{T}_j$: $k_j \in \arg\max_{k\in\mathcal{K}}\left\{\sum_{t\in\mathcal{T}_j}X_t^k\right\}$. Then,

$$w_{t_{j+1}}^{k_j} \geq w_{t_j+1}^{k_j}\exp\left\{\frac{\gamma}{K}\sum_{t_j+1}^{t_{j+1}-1}\hat{X}_t^{k_j}\right\}$$

$$\geq \frac{e\alpha}{K}W_{t_j}\exp\left\{\frac{\gamma}{K}\sum_{t_j+1}^{t_{j+1}-1}\hat{X}_t^{k_j}\right\}$$

$$\geq \frac{\alpha}{K}W_{t_j}\exp\left\{\frac{\gamma}{K}\sum_{t\in\mathcal{T}_j}\hat{X}_t^{k_j}\right\},$$

where the last inequality holds since $\gamma\hat{X}_t^{k_j}/K \leq 1$. Therefore,

$$\log\left(\frac{W_{t_{j+1}}}{W_{t_j}}\right) \geq \log\left(\frac{w_{t_{j+1}}^{k_j}}{W_{t_j}}\right) \geq \log\left(\frac{\alpha}{K}\right) + \frac{\gamma}{K}\sum_{t\in\mathcal{T}_j}X_t^{\pi}. \tag{6}$$

Taking (5) and (6) together, one has

$$\sum_{t\in\mathcal{T}_j}X_t^{\pi} \geq (1-\gamma)\sum_{t\in\mathcal{T}_j}\hat{X}_t^{k_j} - \frac{K\log(K/\alpha)}{\gamma} - (e-2)\frac{\gamma}{K}\sum_{t\in\mathcal{T}_j}\sum_{k=1}^K \hat{X}_t^k - \frac{e\alpha K\,|\mathcal{T}_j|}{\gamma}.$$

Taking expectation with respect to the noisy rewards and the actions of Exp3.S we have:

$$\max_{k\in\mathcal{K}}\left\{\sum_{t\in\mathcal{T}_j}\mu_t^k\right\} - \mathbb{E}\left[\sum_{t\in\mathcal{T}_j}\mu_t^{\pi}\right] \leq \sum_{t\in\mathcal{T}_j}\mu_t^{k_j} + \frac{K\log(K/\alpha)}{\gamma} + (e-2)\frac{\gamma}{K}\sum_{t\in\mathcal{T}_j}\sum_{k=1}^K \mu_t^k$$

$$+ \frac{e\alpha K\,|\mathcal{T}_j|}{\gamma} - (1-\gamma)\sum_{t\in\mathcal{T}_j}\mu_t^{k_j}$$

$$= \gamma\sum_{t\in\mathcal{T}_j}\mu_t^{k_j} + \frac{K\log(K/\alpha)}{\gamma} + (e-2)\frac{\gamma}{K}\sum_{t\in\mathcal{T}_j}\sum_{k=1}^K \mu_t^k + \frac{e\alpha K\,|\mathcal{T}_j|}{\gamma}$$

$$\overset{(a)}{\leq} (e-1)\gamma\,|\mathcal{T}_j| + \frac{K\log(K/\alpha)}{\gamma} + \frac{e\alpha K\,|\mathcal{T}_j|}{\gamma}, \tag{7}$$

for every batch $1 \leq j \leq m$, where (a) holds since $\sum_{t \in \mathcal{T}_j} \mu_t^{k_j} \leq |\mathcal{T}_j|$ and $\sum_{t \in \mathcal{T}_j} \sum_{k=1}^{K} \mu_t^k \leq K |\mathcal{T}_j|$.

**Step 3.** Taking (3) together with (7), and summing over $m = \lceil T/\Delta_T \rceil$ batches we have:

$$
\begin{aligned}
\mathcal{R}^\pi(\mathcal{V}, T) &\leq \sum_{j=1}^{m} \left( (e-1)\gamma |\mathcal{T}_j| + \frac{K \log(K/\alpha)}{\gamma} + \frac{e\alpha K |\mathcal{T}_j|}{\gamma} + 2V_j \Delta_T \right) \\
&\leq (e-1)\gamma T + \frac{e\alpha KT}{\gamma} + \left( \frac{T}{\Delta_T} + 1 \right) \frac{K \log(K/\alpha)}{\gamma} + 2V_T \Delta_T. \qquad (8)
\end{aligned}
$$

Setting the tuning parameters to be $\alpha = \frac{1}{T}$ and $\gamma = \min\left\{ 1, \left( \frac{2V_T K \log(KT)}{(e-1)^2 T} \right)^{1/3} \right\}$, and selecting a batch size $\Delta_T = \left\lceil (\log(KT) K)^{1/3} \cdot \left( \frac{T}{2V_T} \right)^{2/3} \right\rceil$ one has:

$$
\mathcal{R}^\pi(\mathcal{V}, T) \leq 8(e-1)\left( K V_T \log(KT) \right)^{1/3} \cdot T^{2/3}.
$$

Finally, whenever $T$ is unknown, we can use Exp3.S as a subroutine over exponentially increasing pulls epochs $T_\ell = 2^\ell$, $\ell = 0, 1, 2, \ldots$, in a manner which is similar to the one described in Corollary 8.4 in [1] to show that since for any $\ell$ the regret incurred during $T_\ell$ is at most $C\left( K V_T \log(K T_\ell) \right)^{1/3} \cdot T_\ell^{2/3}$ (by tuning $\alpha$ and $\gamma$ according to $T_\ell$ in each epoch $\ell$), and for some absolute constant $\tilde{C}$, we get that $\mathcal{R}^\pi(\mathcal{V}, T) \leq \tilde{C} \left( \log(KT) \right)^{1/3} (K V_T)^{1/3} T^{2/3}$. This concludes the proof. ∎

## C   Numerical Results

**Setup.** We illustrate the upper bound on the regret by a numerical experiment that measures the average regret that is incurred by Rexp3, in the presence of changing environments. We consider instances where two arms are available: $\mathcal{K} = \{1, 2\}$. The reward $X_t^k$ associated with arm $k$ at epoch $t$ has a Bernoulli distribution with a changing expectation $\mu_t^k$:

$$
X_t^k = \begin{cases} 1 & \text{w.p. } \mu_t^k \\ 0 & \text{w.p. } 1 - \mu_t^k \end{cases}
$$

for all $t = 1, \ldots, T$, and for any pulled arm $k \in \mathcal{K}$. The evolution patterns of $\mu_t^k$, $k \in \mathcal{K}$ will be specified below. At each epoch $t \in \mathcal{T}$ the policy selects an arm $k \in \mathcal{K}$. Then, the binary rewards are generated, and $X_t^k$ is observed. The pointwise regret that is incurred at epoch $t$ is $X_t^k - X_t^{k_t^*}$, where $k_t^* = \arg\max_{k \in \mathcal{K}} \mu_t^k$. We note that while the pointwise regret at epoch $t$ is not necessarily positive, its expectation is. Summing over the whole horizon and replicating 20,000 times for each instance of variation, the average regret approximates the expected regret compared to the dynamic oracle.

**First stage (fixed variation, different time horizons).** The objective of the first part of the simulation is to measure the growth rate of the average regret incurred by the policy, as a function of the horizon length, under a fixed variation budget. We use two complementary instances. In the first instance the expected rewards are sinusoidal:

$$
\mu_t^1 = \frac{1}{2} + \frac{1}{2} \sin\left( \frac{V_T \pi t}{T} \right), \qquad \mu_t^2 = \frac{1}{2} + \frac{1}{2} \sin\left( \frac{V_T \pi t}{T} + \pi \right)
$$

for all $t = 1, \ldots, T$. In the second instance similar sinusoidal rewards evolution is "compressed" into the first third of the horizon, where in the rest of the horizon the expected rewards remain fixed:

$$
\mu_t^1 = \begin{cases} \frac{1}{2} + \frac{1}{2} \sin\left( \frac{3V_T \pi t}{T} + \frac{\pi}{2} \right) & \text{if } t < \frac{T}{3} \\ 0 & \text{otherwise} \end{cases} \qquad \mu_t^2 = \begin{cases} \frac{1}{2} + \frac{1}{2} \sin\left( \frac{3V_T \pi t}{T} - \frac{\pi}{2} \right) & \text{if } t < \frac{T}{3} \\ 1 & \text{otherwise} \end{cases}
$$

for all $t = 1, \ldots, T$. Both instances describe different changing environments under the same (fixed) variation budget $V_T = 3$. While in the first instance the variation budget is spent throughout the whole horizon, in the second one the same variation budget is spent only over the first third of the horizon. For different values of $T$ (between 3000 and 40000) and for both variation instances

Figure 1: **Numerical simulation of the performance of Rexp3.** (*Upper left*) The average performance trajectory in the presence of sinusoidal expected rewards, with a fixed variation budget $V_T = 3$. (*Upper right*) The average performance trajectory under an instance in which the same variation budget is "spent" only over the first third of the horizon. In both instances the average performance trajectory of the policy is generated along $T = 5,000$ epochs. (*Bottom*) Log-log plots of the averaged regret as a function of the horizon length $T$.

we estimated the regret through 20,000 replications (the average performance trajectory of Rexp3 for $T = 5000$ is depicted in the upper-left and upper-right plots of Figure 1).

This stage of the simulation illustrates the decision process of the policy, as well as the order $T^{2/3}$ growth rate of the regret. The upper parts of Figure 1 describe the performance trajectory of the policy. One may observe that the policy identifies the arm with the higher expected rewards, and selects it with higher probability, by updating the probabilities of selecting each arm according to the received rewards. While the policy adapts quickly to the changes in the expected rewards (and in the identity of the "better" arm), it keeps experimenting with the sub-optimal arm (the policy's trajectory doesn't reach the one of the dynamic oracle). While Exp3 explores at an order of $\sqrt{\Delta_T}$ epochs in each batch, restarting it every $\Delta_T$ (recall that $V_T$ is fixed, therefore the number of batches is of order $T^{1/3}$, each batch has an order of $T^{2/3}$ epochs) yields an exploration rate of order $T^{2/3}$. The lower parts of Figure 1 show plots of the natural logarithm of the averaged regret as a function of the natural logarithm of the the horizon length. All the standard errors of the data points in these log-log plots are lower than $0.004$. These plots detail the linear dependence between the natural logarithm of the averaged regret, and the natural logarithm of $T$. In both cases the slope of the linear fit for increasing values of $T$ supports the $T^{2/3}$ dependence of the minimax regret.

**Second stage (increasing the variation).** The second stage of the simulation aims to measure how the growth rate of the averaged regret (as a function of $T$) established in the first part changes when the variation increases. For this purpose we used a variation budget of the form $V_T = 3T^{\beta}$. Using first instance of sinusoidal variation, we repeated the first step for different values of $\beta$ between $0$ (implying a constant variation, that was simulated at the first stage) and $1$ (implying linear variation). The upper plots of Figure 2 depicts the average performance trajectories of the Rexp3 policy under different variation budgets. The different slopes, representing different growth rate of the regret for different values of $\beta$ appear in the table and the plot, at the bottom of Figure 2.

This part of the simulation demonstrates the impact of the allowed variation on the policy's decisions. In particular, as $\Delta_T$ is of order $(T/V_T)^{2/3}$, holding $T$ fixed and increasing $V_T$ impacts the batch size. This is illustrated at the top plots of Figure 2. The slope of the linear log-log fit is shown at the bottom of Figure 2, demonstrating the growth rate of the regret when the variation in-

Figure 2: **Variation and performance.** (*Upper left*) The averaged performance trajectory for $V_T = 1$, and $T = 5000$. (*Upper right*) The averaged performance trajectory for $V_T = 10$, and $T = 5000$. (*Bottom*) The slope of the linear fit imply growth rate $V_T^{1/3}$.

creases, supporting the $V_T^{1/3}$ dependence of the minimax regret, and emphasizing the full spectrum of minimax regret rates (of order $V_T^{1/3}T^{2/3}$) obtained under different variation levels.

# References

[1] P. Auer, N. Cesa-Bianchi, Y. Freund, and R. E. Schapire. The non-stochastic multi-armed bandit problem. *SIAM journal of computing*, 32:48–77, 2002.