[Reviews · NeurIPS 2014]

Submitted by Assigned_Reviewer_19

The paper aims at addressing an important problem in the literature of multi-armed bandit, namely whether it is possible to prove sub-linear regret bounds in the case of non-stationary mean rewards with finite variation? It provides a definitive answer for these questions by proving matching lower and upper bound of order O(T^(2/3) ) under the assumption that the total variation of mean-rewards are bounded by some V_T\geq 0 for T rounds. The upper bound is proven for a phased version of EXP3 algorithm which resets the algorithm after every O((T/V_t)^(2/3)) rounds.

General:

The paper is very well-written and easy to follow. I checked the main results and technical proofs (those presented in the main paper) and could not find any error. Also the key passages of the technical proof have been well explained. Overall I believe the paper includes enough contribution for a conference submission.

Few technical comments:

1- A closely related setting to the non-stationary reward scenario is the state-dependent ergodic bandits (e.g., restless bandits) as it is recognized in the paper. Although some of the earlier works in ergodic bandits have been discussed, the paper does not completely cover the-state-of-the-art of this field and misses some of the new works (e.g. see resteless bandit of (Ortner 12) for regret bounds for restless bandits with unknown dynamics and the bandit for correlated feedback of (Azar 14) for history-dependent reward bandits).

2- Rexp3 requires to receive V_T as an input. This may turn out to be a restrictive requirement in problems with unknown variation of rewards. I wonder whether one can relax this requirement and still achieve a same regret? I would like to see some discussion regarding those cases in which V_T under-estimated.

3- The proposed algorithm utilizes a variant of EXP3, which is designed for adversarial bandit, to deal with a problem which is stochastic (albeit time varying) in nature. Although this seems to be sufficient for achieving the minimax rate in the worst-case scenario, it does not exploit the stochastic structure and therefore might not be the best problem dependent strategy. I wonder whether one can achieve problem dependent bounds, which can exploit the structure, e.g., gap between best arm and others by extending stochastic bandit algorithms to this setting? maybe, some extension of UCB which can 'forget'?
Summary: Nice results. Technically strong and rigorous. The case of unknown variation bound needs to be discussed.

Submitted by Assigned_Reviewer_20

Secondary Review: The authors have replied to my concerns in a satisfactory manner. In particular, the problem with proof of Thm 1 can indeed be mitigated by the choice of epsilon suggested by the authors, although the second last inequality in their proposed derivation seems strange (the additional V_t^{2/3} would appear unnecessary) and, after noting that the "first part still holds" they can move directly to the case when \epsilon = 1/4 \sqrt{K/\tilde{\Delta}_T}. About line 424, their meaning is clear, however I see no reason to exclude the T^{1/3} term, as it provides information about the initial performance, and leaving it out all together would be misleading.

Thanks to the authors for correcting their derivations, and I am happy to change my review scores accordingly.

Initial Review:

Summary of paper: The authors consider the multi-armed bandit (MAB) problem in a setting where the rewards are generated by non-stationary distributions. In particular they introduce a quantity ($V_T$ in the paper) which upper bounds the amount that the means of the reward distributions are allowed to vary by. They consider classes of MAB problems defined through admissible ranges for $V_T$, and they measure the worst-case regret of algorithms against a (non-stationary) oracle over these classes. They give a result lower bounding this worst-case regret of any strategy for these problems in terms of $V_T$, the time horizon, and the number of arms, and they give a matching upper bound on an epoch-variant of the Exp3 strategy. Thus they show a tight dependency of this class of problems on the quantity $V_T$, and motivate its interpretation as a quantitative measure of the cost of introducing non-stationarity into the standard MAB problem.

Quality and Clarity: The paper is well written in parts, but not in others. In particular I found it hard to follow the details of the calculations in the proofs, and the many new notations. First two major issues:
- It seems to me that there is a mistake in the proof of Theorem 1. Specifically on line 119 in the appendix I do not understand how the second inequality follows from the assumption that $T\ge KV_T$. Indeed, it seems to me that the numerator on the LHS of this inequality is greater than or equal to $T^{4/3}/8$ iff $(KV_T)/T \ge (1+\sqrt{\log(4/3)}/2)^3/2$, which is not implied by the assumption mentioned. I would appreciate it if the authors could clarify how the argument proceeds at this point.
- I have difficulty in reconstruction the bound (a) on line 424 in the proof of Theorem 2. In particular, using the authors assumption that $V_T\ge K^{-1}$ I can bound the second term on the RHS of the inequality above (a) by $K(\log K)^{2/3}T^{1/3}$. Crucially, here there is a linear dependence on K. I would appreciate it if the authors could clarify how this becomes a dependence on $(KV_T)^{1/3}$. This last derivation is of great importance, as if it does not hold, then the bound is not tight with the lower bound provided in Theorem 1.
I list also a few further suggestions that would have helped me to read the paper more fluently:
- Some notation and notation and nomenclature differed from my experience of related literature, whereas it could be easily brought in line with it. In particular I found the "Admissible policies, performance and regret" paragraph did not introduce policies in a standard way, as they are introduced in many bandit and MDP papers (I think it is more normal to treat \pi as a distribution over the arm set that is measurable with respect to the history, and to denote by a_t, or I_t the chosen arm at time t).
- The two paragraphs after Theorem 2 could be emphasised more since they are important and interesting consequences of the theorem and its proof.
- I do not understand why the proof of Theorem 2 is in an Appendix within the main submission. It would be more natural for it to be a section before the discussion, or that it come immediately after Theorem 2.
- The connection with reference [27] is often alluded to, but not in the end explained in any detail. It seems all of these allusions could therefore be in one place, for example the discussion section.

Originality and significance: The contribution is original, and of significant interest to the NIPS community. It adds new understanding to the interface between stationary and non-stationary MAB problems. This should be of practical interest, although the authors do not directly suggest any real-world applications where the quantity $V_T$ can be interpreted.
Summary: This is an original contribution to the multi-armed bandit literature in which the authors consider settings where the rewards are generated by non-stationary distributions, and give matching upper and lower bounds on a worst-case regret measuring performance against a (non-stationary) oracle.

Extra sentence from the first review, which has been addressed: Unfortunately the paper is not very easy to read and could be better structured; moreover there appear to be two technical problems with the proofs, making it impossible for me to give the paper a score higher than 3.

Submitted by Assigned_Reviewer_31

The authors introduce a new model for multi-armed bandits with non stationary rewards based on the idea of imposing an upper bound on the total variation of changes in the expected rewards of the arms. They prove a lower bound and introduce a new algorithm (that is an extension of EXP3) with a matching upper bound in terms of the dependency on the time horizon.

The paper is well structured and written. The related work is properly cited and the connection with previous papers is reasonably covered.

I think the model is not sufficiently motivated. While setting a maximum bound on the variation of expected rewards is natural from a theoretical standpoint, it is not immediately obvious how one would test such a property in a practical setting. I think the authors should improve the presentation from this perspective. That being said, setting the problem in this way leads to an interesting intermediate setting between stochastic and adversarial bandit settings.

Regarding the lower bound, I enjoyed the presentation of the sketch of the proof for Theorem 1. The ideas are nice and presented succinctly and clearly.

One key issue with the paper from my point of view is the definition of Rexp3. The algorithm is defined to have an optimal bound in the worst case (that from Theorem 1) in the sense of restarting an EXP3 instance at the beginning of every epoch. While it might be enough from a theory standpoint, it seems highly impractical to completely forget everything at the beginning of an epoch. Another reason of concern is that V_T is an input to the algorithm. Why is it reasonable to assume V_T is known or can be learned? I am not claiming it is not possible, but I think the authors should at least discuss this assumption.

I also have a question regarding Theorem 2: the proof holds for V_T >= 1/K (as mentioned in the theorem statement). What happens with the algorithm when V_T gets very close to 0 (the stationary case) and much smaller than 1/K? I understand the theorem doesn’t cover this case, but it seems like a natural scenario to consider.

[Update] I have read the author feedback and I consider it satisfactory so I increase the score to 6: Marginally above the acceptance threshold.
Summary: The paper is well written and tackles an interesting and very relevant problem. At this point, due to the issues with the algorithm described above, I am inclined to recommend rejection.
Author Feedback
Author rebuttal: Reviewer 19
1. Thank you for the additional refs, which we will add in the final version.
2. While the introduction of the variation budget is an important departure from assuming stationary or knowledge of the structure of changes, we agree that the requirement that an upper bound on V_T is known could be restrictive. In the final version we will discuss prior knowledge of V_T in Section 5. It is not clear that one may guarantee sub-linear regret with unknown V_T, unless one places additional restrictions on the type of variation (for example, in [26] considers a MAB problem with no bound on the variation, but instead the number of switches in the identity of the best arm is finite). We will also discuss the consequences of under-estimating V_T. If Rexp3 is tuned by V_T = T^{a}, but the variation is T^{a + delta}, Rexp3 incurs sublinear regret (of order T^{2/3 + a/3 + delta}) as long as delta < 1 - a - 2/3, and linear regret otherwise; e.g., if a=0 and delta = 1/4, Rexp incurs order T^{11/12} (accurate tuning of Rexp3 would have guarantee order T^{3/4}).
We note that given historical data on rewards of all arms (which can be obtained, for example, if actions are occasionally randomized), it is possible to estimate an upper bound on the variation. For example, one may fit V_T = T^{alpha}.
3. It may be possible to adapt policies that are designed for stochastic reward structure (in a manner that accounts for non-stationarity) to achieve good performance in stochastic non-stationary environments. [25] does that but focuses on a more restrictive setting, where changes in the rewards are “large”. There are significant technical difficulties in establishing tight regret bounds for the general variations allowed in the present paper.
This being said, we note that different types of policies may be analyzed (see response to reviewer 31) and that our results imply that, at least in terms of regret rate (as a function of T, V_T, and K), policies that exploit the stochastic structure of the problem cannot outperform Rexp3.

Reviewer 20
Line 119 in the Appendix: There was indeed a typo in the definition of epsilon, however the result holds, and in the revision the proof will be adjusted and simplified as follows. First, select \varepsilon = \min \{(1/4) \sqrt{K / \tilde{\Delta}_T} , V_T \tilde{\Delta}_T / T \}. Note that part 1 still holds when T is large enough. Then, from line 107 one has:
R\geq T \varepsilon (1/2 - \varepsilon \sqrt{\tilde{\Delta}_T} \log(4/3)/\sqrt{K})
\geq T \varepsilon\ (1/2 - \sqrt{\log(4/3)}/4)
\geq (1/4)\min\{(T/4)\sqrt{K/\tilde{\Delta}_{T}} , V_{T}\tilde{\Delta}_{T}\}
\geq (1/4)\min\{(T/4)\sqrt{\frac{KV_{T}^{2/3}}{K^{1/3}T^{2/3}+V_{T}^{2/3}}} , (KV_{T})^{1/3}T^{2/3}\}
\geq (1/(16\sqrt{2}))(K V_T)^{1/3} T^{2/3}.
Line 424 in main text: We apologize for the confusion that arose. In the paper, we aim to develop policies with the best possible regret dependence on T as a first objective and second, we also aim to characterize the impact of the number of arms on the multiplicative constant that may appear in front of the T factor. This is motivated by the fact that in typical situations (and a typical assumption in the MAB literature), K is of lower order than T (o/w it is not clear what one may learn; see, e.g., [Y. Deshpande and A. Montanari, Linear Bandits in High Dimension and Recommendation Systems, 2012] for other objectives). As a result, we consider the dependence on K only in the dominant T terms, i.e., those that include T^{2/3} in the proof. (The term the reviewer was referring to had a T^{1/3} dependence.) We hope that clarifies the result and that there is no mistake. In the final version we will better articulate this.
Notation: We will streamline the notation (following, e.g., [26]).
Paragraphs after Thm 2: We agree, and will emphasize those more in the final version.
Position of the proof of Thm 2: We agree, and will move the proof to the main text, as suggested.
Ref [27]: In [27] rewards change according to a Brownian motion; in the context of our framework V_T is linear in T, and indeed regret is linear. In the final version we will further clarify, as suggested, the relation to [27] in a single place (Section 5).

Reviewer 31
Model motivation: In the final version we will further motivate the variation budget idea. One practical motivation is that given historical data on rewards of all arms (which can be obtained, for example, when actions are occasionally randomized) it is possible to estimate V_T. For example, one may fit V_T=T^{alpha}.
Theory vs practical policies: The purpose of designing the Rexp3 policy is to emphasize the principles one should consider in a changing environment to achieve rate optimality. Rexp3 is only one way of achieving rate-optimality; one may also adapt the Exp3.S policy from [26] to achieve rate-optimality through a policy that adapts weights ``smoothly’’. In the final version we will articulate this point after Theorem 2, indicating how this may be done. The latter is more practical but the former articulates more clearly the principles leading to rate optimality.
We note that the results imply that, at least in terms of regret rate (as a function of T, V_T, and K), policies that address these principles in any other way rather than restarting cannot outperform Rexp3.
Prior knowledge of V_T: Please see point 2 of the response to reviewer 19.
Low variation rates: We agree that other rates of change in V_T are also of interest. The type of analysis conducted can still be applied to obtain parallel results for “small” V_T. When V_T decreases from O(1) to O(T^{-1/2}), the minimax regret decreases from O(T^{2/3}) to O(T^{1/2}), at which point it coincides with the known bounds of the stationary stochastic problem. If V_T is of smaller order than O(T^{-1/2}), the minimax regret is still O(T^{1/2}). In the final version we will articulate and highlight these results.